# Fiber-Reinforced Lightweight Calcium Aluminate Cement-Based Concrete: Effect of Exposure to Elevated Temperatures

Özlem Salli Bideci [1], Hakan Yılmaz [2], Osman Gencel [3,*], Alper Bideci [1], Bekir Çomak [4], Mehrab Nodehi [5] and Togay Ozbakkaloglu [6,*]

[1] Department of Architecture, Faculty of Art, Design and Architecture, Düzce University, 81600 Düzce, Turkey
[2] Institute of Science, Düzce University, 81600 Düzce, Turkey
[3] Department of Civil Engineering, Faculty of Engineering, Bartin University, 74100 Bartin, Turkey
[4] Department of Civil Engineering, Faculty of Technology, Düzce University, 81600 Düzce, Turkey
[5] Department of Civil Engineering, University of California, Davis, CA 95616, USA
[6] Ingram School of Engineering, Texas State University, San Marcos, TX 78666, USA
[*] Correspondence: ogencel@bartin.edu.tr (O.G.); togay.oz@txstate.edu (T.O.)

**Abstract:** Calcium aluminate cements (CACs) are a group of rapid-hardening hydraulic binders with a higher aluminum composition and lower ecological footprint compared to their ordinary Portland cement (CEM) counterparts. CACs are commonly known to have higher thermo-durability properties but have previously been observed to experience a major strength loss over time when exposed to thermal and humidity conditions due to the chemical conversion of their natural hydrated products. To address this, in this study, silica fume is added to induce a different hydration phase path suggested by previous studies and utilized in conjunction with fiber-reinforced lightweight pumice to produce lightweight concrete. To closely evaluate the performance of the produced samples with CAC compared to CEM, two different types of cement (CEM and CAC) with different proportions of pumice and crushed stone aggregate at temperatures between 200 and 1000 °C were tested. In this context, sieve analysis, bulk density, flowability, compressive and flexural strength, ultrasonic pulse velocity and weight loss of the different mixes were determined. The results of this study point to the better mechanical properties of CAC samples produced with pumice aggregates (compared to crushed stone) when samples are exposed to high temperatures. As a result, it is found that CACs perform better than CEM samples with lightweight pumice at elevated temperatures, showing the suitability of producing lightweight thermal-resistant CAC-based concretes.

**Keywords:** calcium aluminate cement (CAC); lightweight concrete; pumice; thermal performance; silica fume

## 1. Introduction

CACs, or high alumina cements, are a group of rapid-hardening, sulfate-resistant cementing materials that are made from limestone and bauxite after being melted in a reverberatory furnace with a higher content of $Al_2O_3$ compared to CEM [1]. Based on the aluminate content, CACs are classified as low (with Al of 36–42%), intermediate (Al of 48–60%) and high (Al > 80%) purity, with a higher aluminum content having higher resistance to the thermo-durability causes of deterioration [2].

Initially, the development of CACs took place during the 1900s, and the first patent was later filed in 1909 by Bied on low silica containing cement under the name "Ciment Alumineux" [3]. The early intention of the development of such cement was as a sulfate-resistant binder, but it was later discovered that it has other major benefits, such as better acid, abrasion and expansion resistance as well as a high strength development rate, even in very low temperatures [3,4]. On this basis, CACs started to be commercialized as early as

1918 under the name Ciment Fondu Lafarge [3,4]. Soon after, CACs became commonly used cements in refractory industries and precast structural sections due to their fast strength gain rate and superior thermo-durability properties [4]. Later in the 1960s, however, it was discovered that CACs tend to go through a deteriorating conversion process, especially when the initial w/b ratio of above 0.4 and cement content of below 400 kg/m$^3$ was used [1], which was a common practice due to the lack of plasticizer availability at the time. This resulted in the failure of certain structures and CACs being banned for use in major structural elements in the 1970s [4].

Later studies discovered that the mentioned conversion tendency was due to the result of the inherent hydration mechanism of CACs whereby the initially hardened metastable products convert to denser stable hydrates due to the lower energy state of the metastable products compared to the dissolved ions [3,5]. In that respect, unlike CEM that has a clinker in the form of $C_3S$ and $C_2S$, CACs are mainly composed of $CaO.Al_2O_3$ (CA), $CaO._2Al_2O_3$ ($CA_2$) and $12CaO.7Al_2O_3$ ($C_{12}A_7$), with the $CA_2$ reported to have high thermal resistance [2,6]. When hydrated, a product consisting of $Al_2O_3 \cdot 3H_2O$ (AH3), $3CaO \cdot Al_2O_3 \cdot 6H_2O$ ($C_3AH_6$), $2CaO \cdot Al_2O_3 \cdot 8H_2O$ ($C_2AH_8$) and $CaO \cdot Al_2O_3 \cdot 10H_2O$ ($CAH_{10}$) is produced [6,7]. Followed by this, the CAH10 and C2AH8 tend to convert to the more stable C3AH6 and AH3, which takes place at higher temperature (e.g., 60 °C) and favorable humidity, that results in increased porosity and an eventual loss of strength [6–9]. Table 1 shows the chemical composition, structural shape and density of the mentioned products. Based on this table, on average, $C_3AH_6$ and $AH_3$ have about 34% higher density values compared to $CAH_{10}$ and $C_2AH_8$.

**Table 1.** Chemical properties of CAC hydrates [10].

| Hydrates | Chemical Composition (%) | | | Structure Shape | Density (g/cm$^3$) |
|---|---|---|---|---|---|
| | CaO | Al$_2$O$_3$ | H$_2$O | | |
| CAH$_{10}$ | 16.6 | 30.1 | 53.3 | Hexagonal | 1.743 |
| C$_2$AH$_8$ | 31.3 | 28.4 | 40.3 | Hexagonal | 1.950 |
| C$_3$AH$_6$ | 44.4 | 27.0 | 28.6 | Cubic | 2.527 |
| AH$_3$ | - | 65.4 | 34.8 | Hexagonal | 2.420 |

To address this common shortcoming, previous studies (e.g., [8,9,11,12]) suggested the addition of silica content to provide an alternative route for hydration-forming C-A-S-H phases instead of the mentioned conversion. As a result, most studies conducted have only focused on this property. For instance, Lee et al. [13] investigated the effect of high temperature exposure of CAC-based ultra-high performance concrete supplied with silica fume. It was documented that dense $C_3AH_6$ continued to be formed resulting from the dehydration of $CAH_{10}$. Nonetheless, an increase in compressive strength was reported due to the formation of C-A-S-H that continued until around 450 °C. In another study, Hidalgo et al. [14] utilized silica fume to reduce the potential leaching of CAC specimens. Despite the reported favorable results due to compaction, only natural aggregate was used, and the physico-mechanical and thermal performance of the mixes were not evaluated. In the same way, Lopez et al. [12] conducted a similar experiment on the microstructural formation of CACs by utilizing a combination of coal fly ash and silica fume to produce ternary binders, but no lightweight aggregate was used. Akcaozoglu [15] used lightweight expanded clay with CAC but only reported the effect of cooling regime on the mechanical properties of the different mixes. Ref. [16] conducted an experiment on the thermal performance of the CACs by exposing samples to a group of temperatures ranging from 13 to 80 °C. In the mentioned study, it was reported that as under higher temperatures certain phase conversations takes place, thermal diffusivity slightly decreases. Similarly, Ref. [17] exposed CAC samples to a temperature range of 20–800 °C and compared the results with Portland cement-made samples. Based on the results, CAC samples performed better than the Portland cement ones, while also experiencing a major loss of 24% after being exposed to 400 °C. This was associated with the conversion reaction. Nonetheless, it was reported

that CAC samples outperform those of Portland cement ones at 400–600 °C, with no major cracks being observed on their surface.

As outlined, most studies have only focused on the conversion tendency of CAC mixes and their resulting performance and have not incorporated lightweight aggregates and fibers that are commonly used in the actual field. On this basis, in this study, silica fume with an average silicon dioxide content of 87.61% was used to increase the silica content of the mixes, as advised by previous studies (e.g., [8,9,11,12]), along with lightweight pumice aggregate. The reason for the addition of silica fume was to follow a different hydration path, as discussed in Refs. [8,9,11,12]. In general, silica fume is a known highly reactive supplementary cementitious material (SCM) with major uses in the production of high-performance and high-strength concretes [18,19].

Pumice was chosen in this study as it is a commonly used lightweight material that is chemically inert and has a good insulation property, making it suitable for the production of lightweight concrete [20,21]. Further, to ensure that the samples have a lowered shrinkage value and resemble field concretes, as recommended in previous studies (e.g., [22,23]), a combination of steel and polypropylene fibers with a quantity of 5 and 0.15% binder wt. was, respectively, added to all of the mixes.

To further evaluate the physico-mechanical and thermo-durability properties of the produced lightweight specimens, six mixes with two different sizes of coarse crushed stone and lightweight pumice (4–8 and 8–16 mm) were used. The results of this study point to the better performance of CAC mixes in producing lightweight structural concrete with enhanced properties, especially at elevated temperatures.

## 2. Materials and Methods

### 2.1. Materials

#### 2.1.1. Aggregate and Filler

Crushed stone and pumice (4–8 and 8–16 mm) with a specific gravity of 2.71 and 2.63 $g/cm^3$ and a water absorption of 2.41 and 14.25 (%) were used as coarse aggregate materials, respectively. In addition, to increase the compaction of specimens, stone powder (0–4 mm) with a specific gravity of 2.69 $g/cm^3$ was added to the mixes. Figure 1 also shows the particle size distribution of the aggregates used in this study, assessed based on ASTM C136 [24]. Further information on the physico-mechanical properties of the aggregates and filler used can be found in Tables 2 and 3. The test results of the water absorption and specific gravity of the aggregates are shown in Table 3. As can be seen in this table, pumice aggregates show a relatively higher water absorption ranging from ~28 to 31% that can provide the internal moisture needed to initiate the conversion mechanism of CAC mortar specimens. The high water absorption of pumice aggregate is due to its internal porosity, as discussed in previous studies [25].

**Table 2.** Chemical analysis of the pumice (wt.%).

| Composition | Content (%) |
| --- | --- |
| $SiO_2$ | 74.10 |
| $Al_2O_3$ | 13.45 |
| $Fe_2O_3$ | 1.40 |
| CaO | 1.17 |
| MgO | 0.35 |
| $K_2O$ | 4.10 |
| $Na_2O$ | 3.70 |
| $SO_3$ | 0.12 |
| Loss on ignition | 1.54 |

**Table 3.** Water absorption ratio and particle density test results of aggregates.

| Aggregates (Size Range in mm) | Water Absorption (%) | Specific Gravity (g/cm$^3$) |
|---|---|---|
| Crushed stone (8–16 mm) | 0.04 | 2.71 |
| Crushed stone (4–8 mm) | 0.04 | 2.71 |
| Pumice (8–16 mm) | 31.3 | 0.96 |
| Pumice (4–8 mm) | 28.5 | 1.05 |
| Natural sand (0–4 mm) | 1.5 | 2.63 |
| Crushed sand | 1.2 | 2.69 |

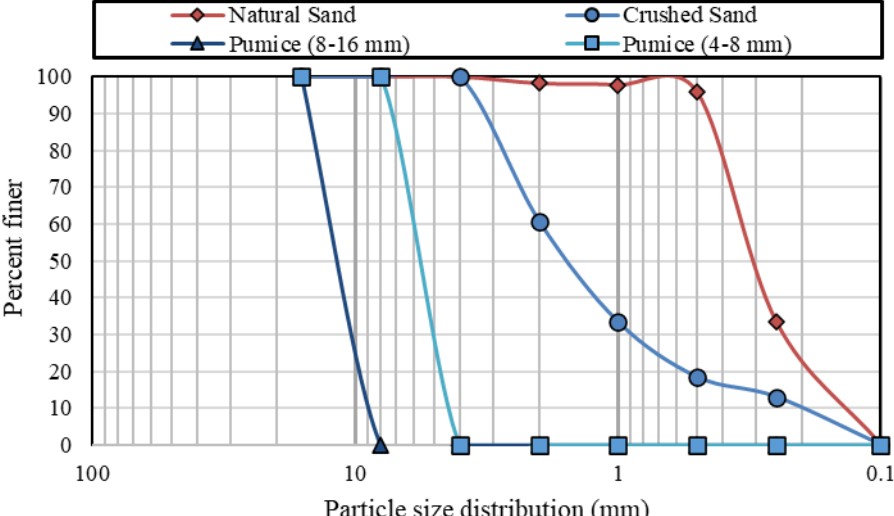

**Figure 1.** Granulometry of aggregate mixture.

### 2.1.2. Cementing Materials

In this study, general purpose CEM-I 42.5 R (CEM) and CAC, with commercial name of ISIDAC 40, containing ~40% $Al_2O_3$ was used. The other physico-mechanical properties of cement materials can be found in Table 4. In addition, silica fume with a constant quantity of 52 kg/m$^3$ was added to all mixes. More information on silica fume is also further presented in Table 4.

**Table 4.** The chemical composition of CEM I 42.5/R and CAC.

| Component (%) | CEM I 42.5R | CAC | Silica Fume |
|---|---|---|---|
| CaO | 63.92 | 36.20 | 0.96 |
| SiO$_2$ | 19.55 | 3.60 | 87.61 |
| Al$_2$O$_3$ | 5.12 | 39.80 | 1.48 |
| Fe$_2$O$_3$ | 2.52 | 17.05 | 2.17 |
| MgO | 1.02 | 0.65 | 1.32 |
| SO$_3$ | 2.96 | 0.04 | 1.95 |
| Na$_2$O | 0.27 | 0.16 | – |
| K$_2$O | 0.67 | 0.11 | – |
| Cl$^-$ | 0.0089 | 0.009 | – |
| Loss on ignition | 4.08 | 0.30 | 3.85 |
| Res. solution | 0.36 | 0.22 | – |
| Specific gravity | – | – | 2.3 |

### 2.1.3. Superplasticizer

To increase the workability of the mixes, polycarboxylic ether-based superplasticizer, with a commercial name of master glenium ACE 450, was added at a constant rate of 2%

by weight of the binder. Further information on the physico-chemical properties of the superplasticizer can be found in Table 5.

**Table 5.** The chemical and physical properties of superplasticizer.

| Type of Superplasticizer | Polycarboxylic Ether Based |
| --- | --- |
| Color | Amber |
| Density | 1.069–1.109 kg/L |
| Chlorine content | <0.1 |
| Alkaline content | <3 |

2.1.4. Fiber

To further enhance mechanical properties, a combination of steel and polypropylene fibers with a quantity of 5 and 0.15% binder wt. was, respectively, added to all of the mixes. In that respect, the steel fiber type was chosen to be hooked wire, with a length of 35 mm, a diameter of 0.7 mm and a tensile strength of 1400 N/mm$^2$, produced according to ASTM A820/A820M-04 [26], which is commercially known as type 1 standard. Table 6 presents the properties of the steel fiber used in further details. In addition, the polypropylene fiber used in study had a length of 12 mm and was produced with specifications conforming to ASTM C-1116 [27].

**Table 6.** Properties of steel fiber.

| Fiber Type | Length L (mm) | Diameter d (mm) | L/d Ratio | Tensile Strength (N/mm$^2$) |
| --- | --- | --- | --- | --- |
| Dramix RC 50/35 BN | 35 | 0.70 | 50 | 1400 |

*2.2. Mix Proportions*

In this study, a total of 6 mixes were used with a constant w/b ratio of 0.36 and two different contents of pumice and natural aggregate with 25 and 100% substitution rate of coarse aggregate. In that respect, natural sand with a size ranging from 0 to 4 mm was used throughout this study to enhance the result of mechanical properties. Further, the mixing IDs used include 3 different sections, including a number denoting the aggregate substitution as a percent, followed by a "P" or "CS" referring to pumice and crushed stone, respectively, plus CEM or CAC. Table 7 shows this specification in more detail, while Table 8 shows the mix design used in this study.

**Table 7.** CEM I 42.5 (CEM) and CAC mixture concrete specimen IDs.

| Mix ID | Information on Mix ID |
| --- | --- |
| 25-PCAC | CAC mixture with 25% pumice aggregate |
| 25-PCEM | CEM-I 42.5 with 25% pumice |
| 100-PCAC | CAC with 100% pumice |
| 100-PCEM | CEM-I 42.5 with 100% pumice |
| 100-CSCAC | CAC with 100% crushed stone |
| 100-CSCEM | CEM-I 42.5 with 100% crushed stone |

**Table 8.** CEM I 42.5 and CAC concrete mixture proportions.

| Mix ID | CEM I 42.5 (kg/m³) | CAC (kg/m³) | Silica Fume (kg/m³) | Pumice 8–16 mm (kg/m³) | Pumice 4–8 mm (kg/m³) | Crushed Stone Filler (kg/m³) | Crushed Stone 8–16 mm (kg/m³) | Crushed Stone 4–8 mm (kg/m³) | Natural Sand 0–4 mm (kg/m³) | Water | Steel Fiber (kg/m³) | Polypropylene Fiber (kg/m³) | Superplasticizer (kg/m³) |
|---|---|---|---|---|---|---|---|---|---|---|---|---|---|
| 100-PCAC | - | 500 | 52 | 154 | 168 | 466 | - | - | 338 | 180 | 25 | 0.75 | 10 |
| 100-PCEM | 500 | - | 52 | 154 | 168 | 466 | - | - | 338 | 180 | 25 | 0.75 | 10 |
| 25-PCAC | - | 500 | 52 | 154 | 168 | 466 | - | 435 | 338 | 180 | 25 | 0.75 | 10 |
| 25-PCEM | 500 | - | 52 | 154 | 168 | 466 | - | 435 | 338 | 180 | 25 | 0.75 | 10 |
| 100-CCAC | - | 500 | 52 | - | - | 470 | 438 | 438 | 340 | 180 | 25 | 0.75 | 10 |
| 100-CCEM | 500 | - | 52 | - | - | - | 438 | 438 | - | 180 | 25 | 0.75 | 10 |

Specimen Preparation and Test Methods

In this study, sieve analysis, specific gravity and water absorption tests based on ASTM C136 [24] and ASTM C127 [28] were conducted on the aggregate materials. Unit weight, dry bulk density and flow table, with a grip and hinge and an overall diameter of 700 mm, based on ASTM C138 [29], ASTM C29 [30], ASTM C230 and EN 12350-5 [31] were, respectively, evaluated. As for compressive and flexural strength test, 54 cubic and prism specimens with size of $100 \times 200$ and $100 \times 100 \times 500$ mm conforming to ASTM C109 [32] and ASTM C348 [33] were, respectively, used and tested after 7, 28 and 56 days of casting. In this experiment, the testing machine used was a uniaxial concrete compression tester with 3000 KN-loading capacity with digital control unit, controlled velocity and fractured in $0.6 \pm 0.2$ Mpa/s constant loading velocity.

The adopted curing in this research was water immersion in which specimens were covered by damp cloth for the initial 24 h and, after demolding, were placed in water with ambient temperature of $20 \pm 2$ °C until tested. Water immersion was specifically used in this research to induce the previously mentioned conversion tendency of CACs, as suggested in previous studies [8,9,11,12]. In this regard, to expose the specimens to thermal stress, total of 90 cylindrical specimens with size of $\varnothing 100 \times 200$ mm were prepared. After removing the specimens from the curing chamber, specimens were stored at room temperature for 24 h. The specimens were then subjected to 200, 400, 600, 800 and 1000 °C in a $310 \times 315 \times 260$ mm temperature-controlled laboratory furnace, with a maximum heating capacity of 1000 °C and temperature increase rate of 10 °C/min. The specimens were kept in furnace for about 1 h until they reached the mentioned temperatures. The specimens, which were kept in for the specified time, were air cooled at room temperature for 24 h and then they were removed from the high temperature furnace and tested for compressive strength, ultrasonic pulse velocity (conforming to ASTM C597 [34]) and weight loss.

## 3. Test Results and Discussions

### 3.1. Flowability

The results of the flow table test (flowability) and unit weight test are shown in Table 9. As can be seen in this table, the substitution of 25% crushed aggregate using pumice can increase the flow value and reduce the unit weight values by about 15 and 11% in CEM-based mixes (100-CSCEM versus 25-PCEM) and 16 and 10% in CAC-based mixes (100-CSCAC versus 25-PCAC), respectively. It can also be seen that when a higher pumice content is used, the flowability of the mixes reduce below the control specimens. This can be due to the high porosity of pumice aggregates, as discussed by previous studies (e.g., [35,36]). In that respect, based on Table 9, when pumice substitutes the crushed aggregate by 100% volume, it results in a mean reduction of the flow table values of 15 and 16.6% in CEM- and CAC-based mixes. The higher reduction in the flowability values in CAC mixes can potentially be due to the faster consumption of water and the subsequent formation of hydrated materials [36].

**Table 9.** Flow table and unit weight values of fresh concrete.

| Specimens | Flowability (mm) |
|-----------|------------------|
| 100-PCAC | 500 |
| 100-PCEM | 550 |
| 25-PCAC | 700 |
| 25-PCEM | 750 |
| 100-CSCAC | 600 |
| 100-CSCEM | 650 |

### 3.2. Density

The dry bulk density and unit weight test results of the concrete specimens are shown in Table 10. As can be seen in this table, the highest and lowest dry bulk density is found to be 1656 (100-PCEM) and 2376 kg/m$^3$ (100-CSCEM), respectively. According to this table,

the results show a 26 and 30% decrease in the values of dry bulk density when pumice fully substituted the crushed aggregates in the CAC- and CEM-based mixes, respectively. Further in Table 10, the result of the unit weight test is also provided. The results show that the substitution of pumice consistently reduces the unit weight values ranging from 1752 (with 100% pumice) to 2430 (0% pumice) for CAC-based samples and 1826 (100% pumice) to 2456 kg/m$^3$ (0% pumice) for CEM mixes. The reason for this can be the lower mass values of pumice compared to the crush stone used in this study.

**Table 10.** Dry bulk density test results.

| Specimens | Dry Bulk Density (kg/m$^3$) | Unit Weight (kg/m$^3$) |
|---|---|---|
| 100-PCAC | 1745 | 1752 |
| 100-PCEM | 1656 | 1826 |
| 25-PCAC | 2060 | 2128 |
| 25-PCEM | 2123 | 2202 |
| 100-CSCAC | 2368 | 2430 |
| 100-CSCEM | 2376 | 2456 |

Figure 2 shows the compressive strength to bulk density ratio of the tested specimens. The values outlined present the performance of each binder compared with each other. Based on this figure, on average, CEM-based samples show a higher compressive strength to bulk density ratio, especially in longer curing periods. In contrast, CAC specimens appear to have a rather steady compressive strength to bulk density ratio throughout the curing period. This can be due to the higher strength gain rate of CAC mixes compared to their CEM companions.

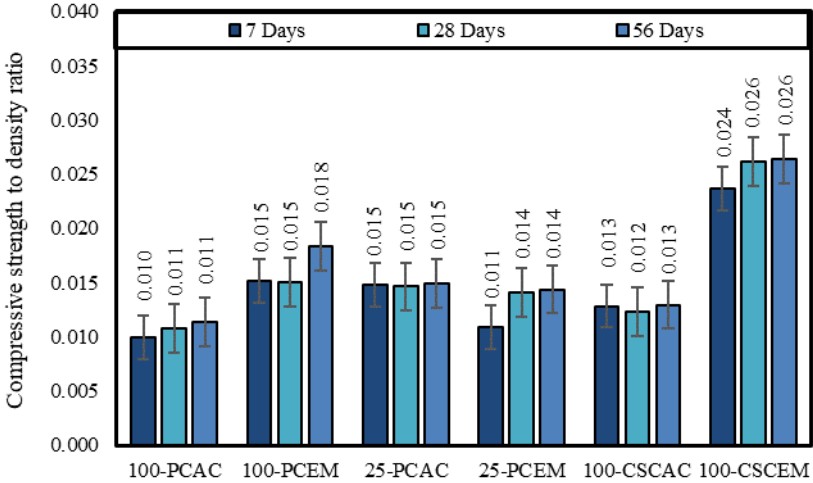

**Figure 2.** Compressive strength to dry bulk density ratio.

### 3.3. Compressive and Flexural Strength

The graph of the compressive strength test results is shown in Figure 3. According to this figure, the CAC mixes develop most of their strength in the initial 7 days of curing. CEM mixes, however, continue to develop compressive strength until the 56th day of measured curing. This shows the slower but steadier compressive strength development of CEM-based samples, compared to the CAC ones, as reported by previous studies [17]. As a result, and based on Figure 3a, the highest compressive strength belongs to 100-CSCEM after 56 days of curing with 62.7 Mpa. Based on this figure, on average, CAC specimens develop 52% lower compressive strength after 56 days of curing compared to their CEM companions. In addition, it can be seen that in the CEM with 25% pumice (25-PSCEM) at 56 days of curing, the compressive strength is reduced by ~51%, while the CAC mix (25-PCAC) experienced no compressive strength loss compared to the 0% pumice mix companion.

However, when pumice is used at 100%, 100-PCAC and 100-PCEM experienced about a 35.5 and 0.6% strength reduction after 56 days of curing. This shows that CAC is more effective in producing lightweight concrete when the lightweight aggregate is used in lower quantities.

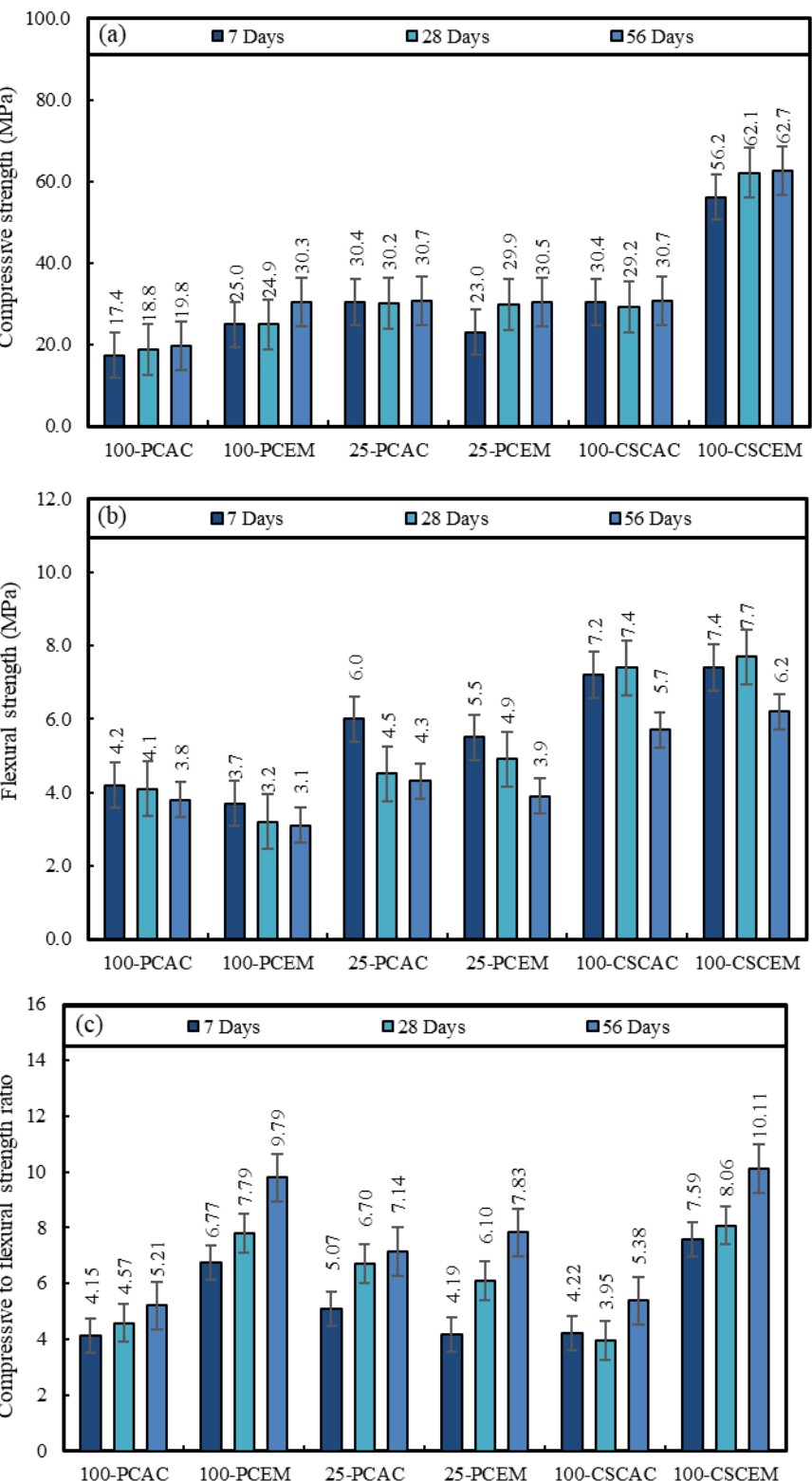

**Figure 3.** Compressive (**a**) flexural strength (**b**) and compressive to flexural strength ratio (**c**) of concrete specimens.

The result of the flexural strength test is illustrated in Figure 3b. Based on this figure, the flexural strength of CEM and CAC mixes ranges from 3.1 to 7.7 MPa and from 3.8 to 7.4, respectively. This points to the better flexural strength of CAC mixes, especially when they contain 100% coarse pumice. Based on this figure, almost all CAC mixes experienced a steady reduction in their flexural strength as the curing duration increased. This can be due to the mentioned microstructural conversion of the matrix, which is aligned with the results reported by Ref. [37].

Figure 3c shows the compressive to flexural strength ratio of the mixes. Based on this figure, CAC mixes have an average of a 47% lower ratio compared to their CEM companions. This shows the better performance of CAC mixes in flexure, while CEM mixes appear to perform best in compression.

### 3.4. Thermal Resistance

The high temperature compressive strength performance of specimens is shown in Figure 4a. Based on this figure, the lowest compressive strength of CAC and CEM specimens is 2.3 and 1.9 MPa that takes place at 1000 °C for mixes containing 25% of pumice aggregate. As discussed by Ref. [38], the addition of pumice increases the amount of physically absorbed water that evaporates at elevated temperatures. This causes higher pore pressure that induces the formation of thermally induced cracks.

The ratio of compressive strength degradation under high temperatures is represented in Figure 4b. Based on this figure, on average, CAC mixes experienced an increase in strength of ~0.4% when subjected to 200 °C and a decline of ~2.0, 40, 71 and 87% when subjected to 400, 600, 800 and 1000 °C, respectively. In addition, as can be seen in this figure, the CEM mixes experienced an average strength decline of 3.5, 17.7, 49.7, 79.7 and 93.7% when subjected to 200, 400, 600, 800 and 1000 °C, respectively. The average total strength loss of CAC and CEM specimens exposed to all elevated temperatures is also found to be ~40 and 48%, respectively. However, at 400 °C, 25-PCAC samples are found to have slightly higher strength values than when exposed to 200 °C. The reason for this can be the chemical conversion of hydrated and un-hydrated particles within the sample that has resulted in a slight variation of the compressive strength values. Further from Figure 4b, it can be seen that CAC mixes appear to experience a major drop in their compressive strength at around and after 400 to 600 °C. This can be due to the loss of the chemically bound water and a major acceleration in the conversion of $CAH_{10}$ and $C_2AH_8$ to $C_3AH_6$, as reported by Refs. [17,39].

In general, however, although all mixes experienced major strength loss at higher temperatures, CAC mixes are found to show a better performance under thermal stress. Figure 4c shows the mass loss of mixes at high temperatures. Based on this figure, at higher temperatures (e.g., 800–1000 °C), CAC samples experience a relatively lower mass loss, compared to their CEM companions. According to Ref. [10], this is due to the mentioned conversion and the fact that the produced $C_3AH_6$ and $AH_3$ do not fully disintegrate until after 1000 °C. Based on this figure, the weight loss of specimens containing pumice under 200 °C is higher than mixes with only crushed stone aggregates (100-CSCAC and 100-CSCEM). This finding is aligned with the result of previous studies such as Ref. [38]. In addition, the largest weight loss of specimens (~31 and 40%) belongs to 100-PCAC and 100-PCCEM containing 100% coarse pumice aggregates. Based on the mentioned, it can be seen that at 1000 °C samples containing both crushed stone and pumice have experienced a lower weight loss than other samples with only crushed stone or pumice.

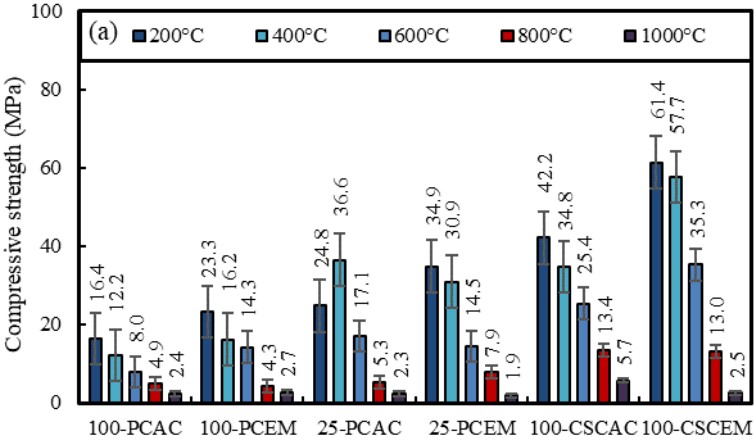

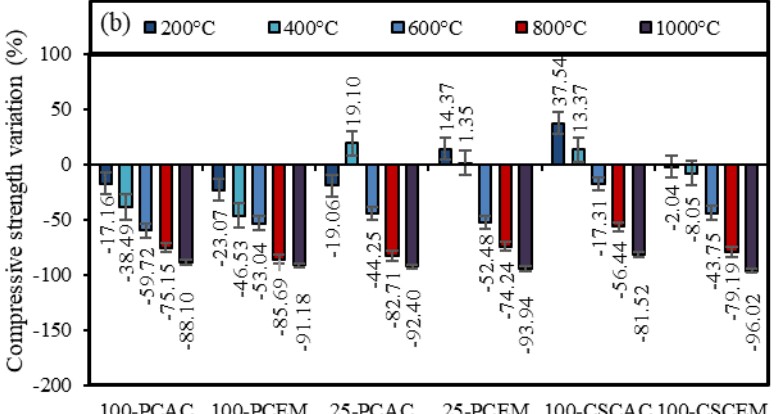

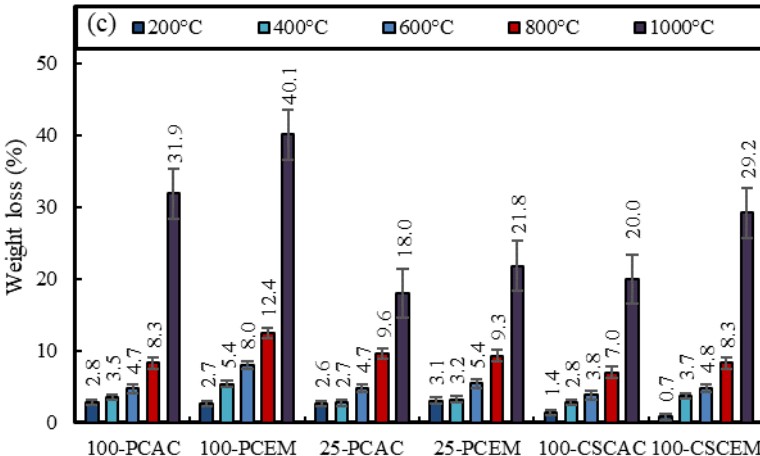

**Figure 4.** Compressive strength: (**a**) compressive strength reduction ratio (**b**) and weight loss (**c**) of samples exposed to high temperatures.

To further evaluate the effect of high temperatures on the internal and microstructural integration of specimens, UPV was conducted, and its results are presented in Table 11. The percent variation of UPV results before and after the test is also shown in Figure 5. Based on this information, specimens containing pumice aggregate are found to experience a larger reduction in their UPV speed. According to Ref. [40], this can be due to the evaporation of water trapped in the aggregate pores that results in a larger reduction of UPV speed. It is worth noting, however, that the ultrasonic velocity of 100-CSCEM after 200 °C is greater

than that before 200 °C, which, in this case, can be due to the formation of ceramic-type bonding from the reaction of $CA_2$ with $Al_2O_3$ to form $CA_6$ [17].

**Table 11.** Ultrasonic pulse velocity (km/h) values of concrete specimens.

| Specimen ID | 200 °C | | 400 °C | | 600 °C | | 800 °C | | 1000 °C | |
|---|---|---|---|---|---|---|---|---|---|---|
| | Before | After | Before | After | Before | After | Before | After | Before | After |
| 100-PCAC | 3.83 | 3.45 | 4.11 | 2.93 | 3.83 | 1.56 | 3.39 | 0.18 | 3.9 | 0.15 |
| 100-PCEM | 4.03 | 3.82 | 3.86 | 2.12 | 3.96 | 1.49 | 4.16 | 0.38 | 4.11 | 0.14 |
| 25-PCAC | 4.59 | 4.24 | 3.57 | 3.74 | 3.96 | 0.91 | 4.27 | 0.62 | 4.56 | 0.73 |
| 25 PCEM | 4.82 | 3.96 | 3.97 | 3.76 | 3.81 | 0.82 | 3.67 | 0.38 | 4.95 | 0.39 |
| 100-CSCAC | 4.67 | 4.41 | 3.98 | 2.95 | 3.81 | 2.42 | 5.23 | 0.74 | 3.52 | 0.71 |
| 100-CSCEM | 4.45 | 4.56 | 5.12 | 3.79 | 4.6 | 2.82 | 4.65 | 0.54 | 4.14 | 0.34 |

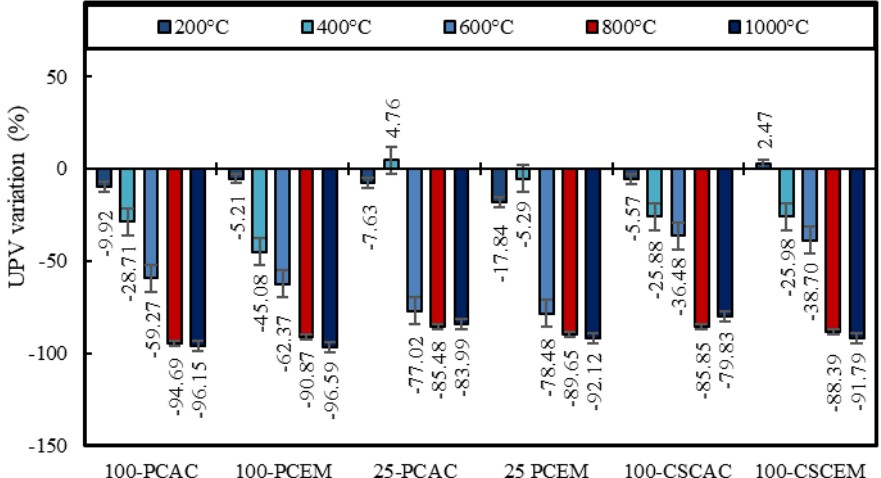

**Figure 5.** UPV variation of heated specimens in percent.

Further, based on Table 11 and Figure 5, on average, the UPV speed reduction rate of CAC and CEM specimens is about −51 and −55%, respectively. In addition, the mixes with 100% pumice substituting the crushed stone content experienced an average of ~58% UPV speed reduction, while specimens containing 100% crushed stone experienced about a ~47% UPV speed reduction. These results are in agreement with earlier reports [41,42].

Figures 6 and 7 show the specimens after being subjected to 800 and 1000 °C temperatures. Based on the observations during this research and these figures, at around 800 °C exposure, the specimens developed considerable cracks and surface spalling. The length of cracks which appeared on pumice-dominated specimens were found to be larger. At 1000 °C, paste disintegration took place in almost all specimens to different degrees. Yet the CEM mixes were observed to experience a relatively higher degree of paste disintegration compared to their CAC counterparts. This observation is aligned with the result of previous studies such as [17,43]. Based on this and the previously reported information [44,45], Figure 8 represents the three common stages of degradation that CAC and CEM are believed to experience. Based on this figure, CEM mixes undergo full microstructural disintegration at around 800 °C, while CAC specimens show a better performance due to the on-going conversion process.

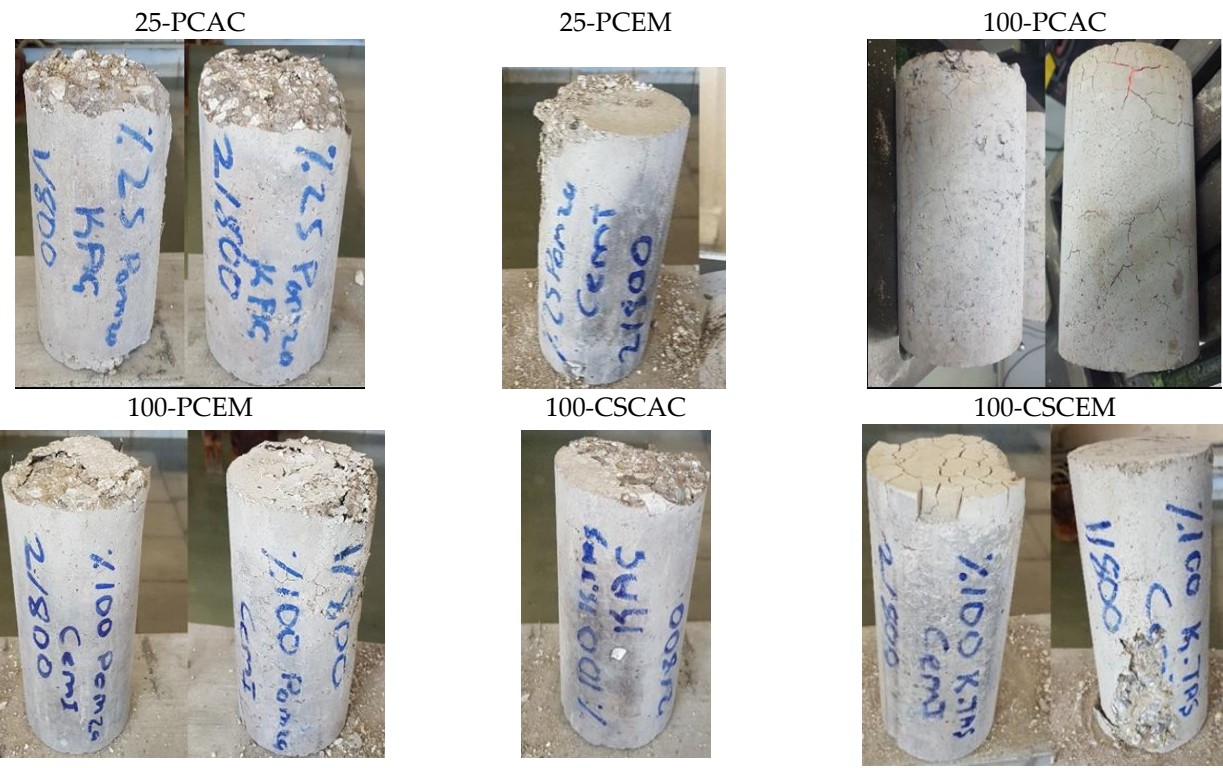

**Figure 6.** Appearance of concrete specimens after 800 °C temperature exposure.

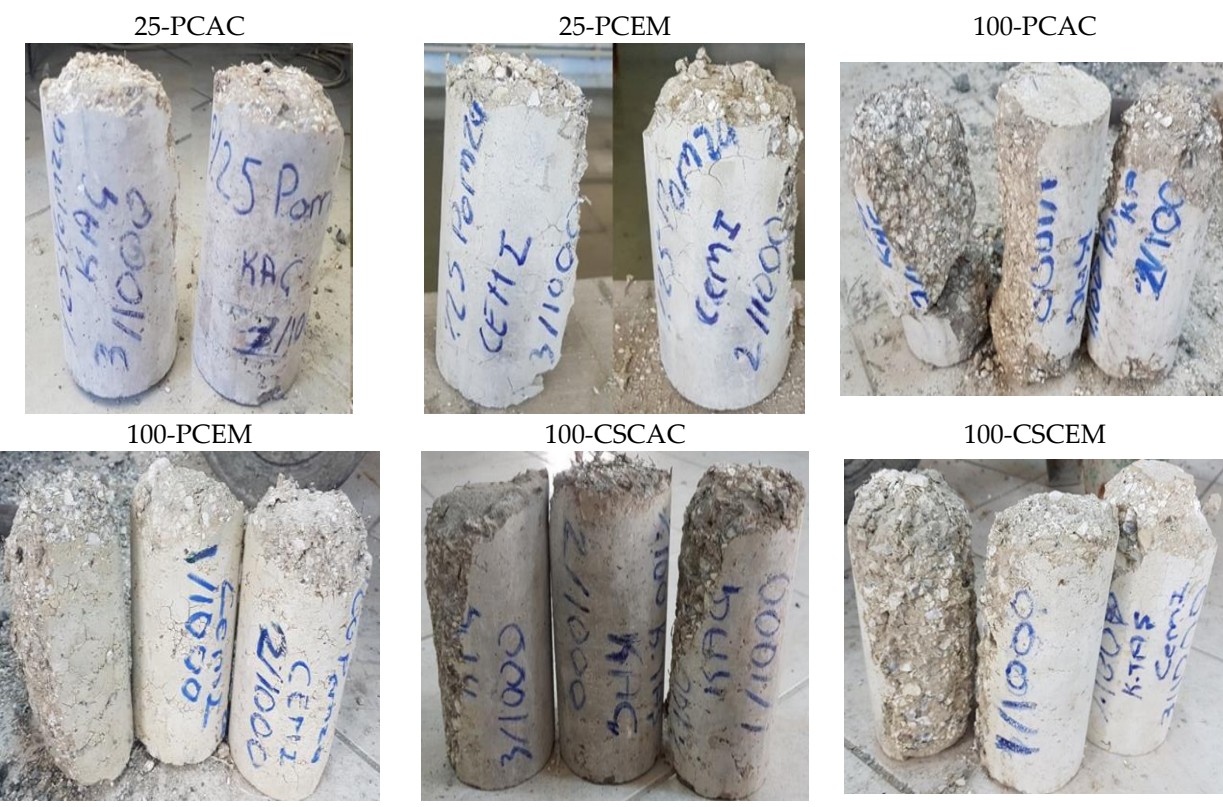

**Figure 7.** Appearance of concrete specimens after 1000 °C temperature exposure.

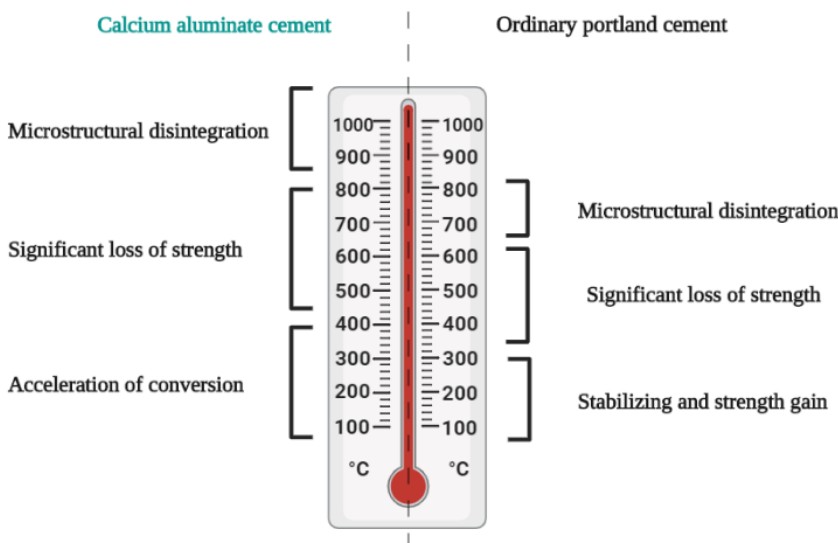

**Figure 8.** The three common stages of degradation that CAC and CEM mixes experienced.

## 4. Conclusions

In this study, lightweight CAC and CEM concrete were made by using a different combination of coarse lightweight pumice and crushed stone. A series of physicomechanical and elevated temperature tests were conducted, and the main results are summarized below:

- The addition of pumice sand is found to increase the flowability of mixes when added up to 25% of the total coarse aggregate content. In larger quantities, however, it is found to reduce the overall flowability of concrete mixes. In terms of unit weight, however, a linear relationship can be made to show the reduction in unit weight as the pumice content increases.

- Based on the result of compressive and flexural strength tests, it is found that CEM mixes have a slower but steadier strength development rate. CAC specimens are found to perform better in terms of flexural strength but experience major flexural strength reduction over time due to the conversion process as a result of microstructural development.

- In high temperatures, surface spalling and cracking is found to take place in almost all the mixes, but CAC samples experience less overall disintegration compared to their CEM counterparts. In that respect, it is found that mixes that contain higher pumice aggregate content experience larger mass loss than the crushed stone-dominated mixes. The same results are also documented during the UPV tests; crushed stone-containing specimens experience less UPV speed reduction compared to their pumice-dominated counterparts. Yet, despite this, pumice-dominated CAC mixes outperform the CEM mixes in elevated temperatures, which shows the better suitability of utilizing lightweight aggregates with CACs.

- Observing samples after being exposed to elevated temperatures, it is found that CEM mixes experience a higher number of cracks and surface spalling. The same tendency is documented for pumice-dominated specimens compared to their 100% crushed stone-containing counterparts.

As recommendations for future studies, the impact of hybrid cementitious usage (e.g., CAC-CEM and CAC–alkali-activated) and their respective microstructural development under high temperatures can be studied.

**Author Contributions:** Conceptualization, O.G. and T.O.; data curation, O.G., Ö.S.B., H.Y., A.B. and B.Ç.; formal analysis, O.G., Ö.S.B., H.Y., A.B., B.Ç. and T.O.; investigation, Ö.S.B., H.Y., A.B. and B.Ç.; project administration, O.G. and T.O.; resources, O.G., Ö.S.B., H.Y., A.B., B.Ç. and T.O.; supervision, O.G. and T.O.; validation, M.N.; visualization, M.N.; writing—original draft, M.N.; writing—review and editing, O.G., M.N. and T.O. All authors have read and agreed to the published version of the manuscript.

**Funding:** This research received no external funding.

**Institutional Review Board Statement:** Not applicable.

**Informed Consent Statement:** Not applicable.

**Data Availability Statement:** Not applicable.

**Acknowledgments:** The authors appreciate all the universities that supported this study.

**Conflicts of Interest:** The authors declare no conflict of interest.

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
