# Peer review of "Fiber-Reinforced Lightweight Calcium Aluminate Cement-Based Concrete: Effect of Exposure to Elevated Temperatures"

_sustainability, doi:10.3390/su15064722_

Round 1
Reviewer 1 Report
The views are as follows:
1. Abstract,“In this context, sieve analysis, bulk density, flow table, compressive and flexural strength, ultrasonic pulse velocity and weight loss of different mixes were determined.”,“flow table” is inappropriate.
2. Introduction, the literature review is insufficient, and the research on the thermal performance of lightweight aggregate concrete should be added in. By the way, the novel of the paper should be highlight.
3. Material and Method, the performance parameters of PP fiber should be supplemented, and it is suggested to add the test results of aggregate (3.1) into the Chapter 2.
4. Test results and discussions, in 3.2 Flowability, there is a logical problem for the statement “Based on this table, the substitution of pumice consistently reduces the unit weight values ranging from 1752 (with 100% pumice) to 2430 (0% pumice) and 1826 (100% pumice) to 2456 kg/m3 (0% pumice) for CAC and CEM mixes”. There is lack of reasonable analysis for the test results, for instance 3.2 Flowability Paragraph 2 (Table 11). In addition, it is recommended that section 3.3 and 3.4 be merged into one section.
5. For 3.4 Compressive and flexural strength, in Fig.3 (b), the flexural strength of CEM mixture decreases with the increase of age, but the alumina content in CEM is not high. Please explain the reason in detail. Additionally, the view“This shows the better performance of CAC mixes in flexure while CEM mixes appear to perform best in compression” cannot be reasonably drawn from the data of compressive to flexural strength ratio.
6. For 3.5 Thermal resistance, in Fig.4. (a), the compressive strength of 25-PCAC at 400 °C is higher than that at 200 °C. For Table 13, the ultrasonic velocity of 100-CSCEM after 200 °C is greater than that before 200 °C. In Fig.4. (c),100-CSCAC and 100-CSCEM may have the lowest weight loss. Please explain reasons for the above in detail.
7. The microstructure morphology of the specimens under different temperature conditions cannot be characterized efficiently by the UPV test results, so extra microscopic test results should be supplemented, such as scanning electron microscopy;
8. There are format problems in this paper, such as the inconsistent text size (Introduction Paragraph 3). And in Table 6, the content “Chlorine content <0.1” seems to be wrong. Major revisions are needed to improve the rationality of the paper.
Reviewer 2 Report
The authors conducted experimental study to invesigate CAC exposed to elevated temprature. The paper is good but needs to improved before acceptance.
This is very obvious results, no need in abstract: "The results of this study point to the better mechanical 24 properties of CEM mixes containing crushed stone while having an inferior thermal property, espe-25 cially when pumice aggregates have been used." Instead of this, abstract should contain importance results in terms of %
The novelty statement should improved.
The authors used fibers but did not mention in abstract and introduction.
The authors should include two paragraphs in introduction.
One of them should be related to silica fume
The second one should be related to the use of steel fibers.
The following should be added to pragraph related to steel fibers: experimental and numerical investigations of steel fiber reinforced concrete dapped-end purlins; improvement in bending performance of reinforced concrete beams produced with waste lathe scraps; performance assessment of fiber-reinforced concrete produced with waste lathe fibers; performance evaluation of fiber-reinforced concretes produced with steel fibers extracted from waste tire; performance evaluation of fiber-reinforced concretes produced with steel fibers extracted from waste tire;
Give the content of fibers in terms of Vf or Wf.
More comments should be added to results f compressive and flexural strength
Please add recommendations for the use of obtained results for engineers
Future studies can be included.
Reviewer 3 Report
sustainability-2147999
Article title:
Self-compacting lightweight calcium aluminate cement (CAC) concrete: Effect of exposure to elevated temperatures
Comments:
The main aim of this work was focused on the effect of CAC cement for LW concrete exposed to elevated temperatures. Great efforts have been performed, but some missing data, typos and corrections need to be clarified.
1. Title: with the title “Self-compacting lightweight calcium aluminate cement (CAC) concrete: Effect of exposure to elevated temperatures”, misleading were found. No related results of details on, for example, ‘self-compacting’, ‘Lightweight aggregate using pumice’,‘Cement or Concrete’ etc. Please revise.
2. Title: “..lightweight calcium aluminate cement (CAC) concrete…” may be revised to something like “..lightweight concrete using CAC and pumice as aggregate…”. Please clarify using ‘Cement’ and ‘Concrete’.
3. Keywords: Some of these keywords may need to be included; Self-compacting, Silica fume, reinforced Fibre, and Pumice.
4. Table 3: Recheck the % absorption of sand with table 10. Is this the same?. And please additionally explain how to test the ‘methylene blue value’ and the meaning of the results.
5. Tables 4&5: and other tables can be combined to shorten the manuscript. It seems to be too many tables.
6. Table 8: Please move the last column (Mix ID) to the first.
7. Section 2.2.1: What is the temperature gradient of the temp-controlled furnace? How about other specifications of it?
8. Section 3.2: Please clarify the words ‘flow table’, ' flow test’, and ‘flow value (result)’.
9. Fig 6&7: Please readjust the portion of the figures.
10. Conclusion: The microstructural development was mentioned, but there were no testing results of, e.g. SEM, XRD or others. Please discuss and clarify this.
11. Overall,
· Why did the author mention “Selt compaction”?
· Discuss using Pumice and show the aim of studying this lightweight concrete.
· How did the added fibre act? What is the control value?
· How did the added Silica fume act? What is the control value?
Reviewer 4 Report
The manuscript titled on “Self-compacting lightweight calcium aluminate cement concrete (CAC): Effect of exposure to elevated temperatures” offers knowledge for using pumice in lightweight CAC concrete. Major revision is required to address as follows:
Title: remove “CAC”
Abstract (Line 20-22): To closely evaluate the performance of the produced CACs, two different types of cement (CEM and CAC) with different proportions of pumice and crushed stone aggregate at temperatures between 200 to 1000 °C have been tested. . Please change CAC compared to CEM
Line 57-61: In that respect, unlike CEM that has a clinker in form of C3S and C2S, CACs are mainly composed of CaO.Al2O3 (CA), CaO.2Al2O3 (CA2) and 12CaO.7Al2O3 (C12A7) with the CA2 having been reported to have high thermal resistance [2,5]. When hydrated, a product consisting of Al2O3·3H2O (AH3), 3CaO·Al2O3·6H2O (C3AH6), 2CaO·Al2O3·8H2O (C2AH8) and CaO·Al2O3·10H2O (CAH10) is produced… the temperature during this hydration should be specified.
In the abstract: The authors provided CAC as a lower carbon footprint. However, in the context, there is no studies addressed. This issue needs to be discussed. In fact, from my review of literature, CAC seems offer higher carbon footprint than CEM.
Line 54-57: The authors used “Later studies discovered…” but the reference is only one reference. Please add more reference as suggested: https://doi.org/10.1016/j.oceram.2022.100290; https://doi.org/10.1080/21650373.2018.1558132
Figure 1: add the particle size of pumice. Or the legions have to change from crushed stone to pumice?
Tables 2, 4, and 5: combine this XRD analyzes into one table
Line 141-142: It is recommended to add PP fiber’s specification into Table 7. And discuss in detail why the fibers are needed in these mixtures.
Table 8: Use CAC instead of “Calcium Aluminate Cemented”
Table 8: the arrangement of each mixture requires revision. The orders of mixtures are not similar with the results, leading to confusion.
Table 9: the column of the mixture proportions should cement and then aggregates.
Line 164: the study assessed flowability using flow table regarding ASTM C230 Standard Specification for Flow Table for Use in Tests of Hydraulic Cement. It is for the CEM mortar, not for the concrete or SCC. The proper standard test should be ASTMC1611: Standard Test Method for Slump Flow of Self-Consolidating Concrete
Begin Line 168-169: As aforementioned in the introduction. CAC is used for rapid-set applications. It seems that the properties at early-age (like 1- or 3-day compressive strength) are needed here. It directly reflects the key properties to its application.
Table 11: The flow value of each specimen is varied. The discussion on how to deal with segregation, especially for mixtures with high flow values such as 25-PCEM and 25-PCAC should be addressed.
Line 221-223: Discuss regarding the reason of assessing compressive strength to bulk density ratio.
Figure 4: The authors detailed on flexural strength to bulk density ratio. The actual values of the flexural strength should be added.
Figure 8: The figure is well-depicted. The images of mechanism
Round 2
Reviewer 1 Report
The views are as follows:
1. Abstract, line 24 to 26, “The results of this study point to the better mechanical properties of pumice aggregates”, pumice aggregates may not be the key of the study, and please revise it.
2. Introduction, the content of carbon footprint for CAC should be added.
3. Material and Method, Table 6, fiber volume is an important factor for fiber reinforced concrete, so the physical properties of fibers (both steel and polypropylene fibers) should be supplied. And the content of fibers is expressed by volume fraction in Table 8.
4. Conclusions, line 373 to 376, this conclusion is not convincing due to the lack of microscopic tests.
5. Please check the paper and correct seriously. For instance, line 304 and 305, “400 C 200 C”, line 223, “3.2 Flowability”, ect.
6. Line 325 to 328, the enhanced compaction is misunderstanding, and there are not sufficient tests to agree this view point.
7. Line 354, Figure 8, it is necessary to explain the conversion from the mechanism perspective. Additionally, the max value of the temperature is 1000 ℃, and in Figure 8 the temperature is higher than 1000 ℃. Recommand to revise the figure to get well with the test results.
Reviewer 2 Report
The paper can be accepted in this current form.
Reviewer 4 Report
The revised manuscript titled on: "Self-compacting lightweight calcium aluminate cement concrete (CAC): Effect of exposure to elevated temperatures" is currently in a good version for publishing in Sustainablilty journal. The authors addressed important information. There are still some typos that need to corrected.
Round 3
Reviewer 1 Report
1 The CAC mixture incorporating with fibre and pumice aggregate is the novelty of this paper. The conversion tendency is the phenomenon for CAC mixture. Although there is related literature to approve the microstructure conversion (eg: line 277-279), there is still lack of tests to approve the results for fiber reinforced lightweight CAC concrete. By the way, the mechanical properties of sample are linking to the microstructure and hydration products. So, it is necessary to add the test results of microstructure.
2 The fiber dose was determined by the corresponding literature or pre-tests. Please elucidate this.
3 Mixing process is vital to fibre reinforced concrete, and sample preparation details should be supplied to improve the test reproducibility.
